# Pharmacology and Emerging Therapies for Group 3 Pulmonary Hypertension Due to Chronic Lung Disease

**DOI:** 10.3390/ph16030418

**Published:** 2023-03-09

**Authors:** Janae Gonzales, Dustin R. Fraidenburg

**Affiliations:** Department of Medicine, University of Illinois at Chicago, Chicago, IL 60612, USA

**Keywords:** pulmonary hypertension, interstitial lung disease, chronic obstructive pulmonary disease, vascular remodeling, hypoxia

## Abstract

Pulmonary hypertension (PH) frequently complicates chronic lung disease and is associated with high morbidity and poor outcomes. Individuals with interstitial lung disease and chronic obstructive pulmonary disease develop PH due to structural changes associated with the destruction of lung parenchyma and vasculature with concurrent vasoconstriction and pulmonary vascular remodeling similar to what is observed in idiopathic pulmonary arterial hypertension (PAH). Treatment for PH due to chronic lung disease is largely supportive and therapies specific to PAH have had minimal success in this population with exception of the recently FDA-approved inhaled prostacyclin analogue treprostinil. Given the significant disease burden of PH due to chronic lung diseases and its associated mortality, a great need exists for improved understanding of molecular mechanisms leading to vascular remodeling in this population. This review will discuss the current understanding of pathophysiology and emerging therapeutic targets and potential pharmaceuticals.

## 1. Introduction

Pulmonary hypertension (PH) is a progressive disease characterized by increased pulmonary vascular resistance and high pulmonary artery pressures that ultimately leads to right heart failure and is associated with high morbidity. PH can occur de novo as either idiopathic or hereditary PAH, but often complicates other chronic conditions including chronic lung diseases. This entity of PH belongs to the Group 3 classification and is one of the highest leading causes of PH worldwide, second to PH associated with left-sided heart disease (Group 2) [1]. Of the several chronic lung diseases, PH is most likely to develop as a complication of interstitial lung disease (ILD) and chronic obstructive pulmonary disease (COPD). Amongst these individuals, the development of PH is associated with increased mortality, reduced functional capacity, and poor quality of life [2,3,4,5,6]. Despite the increased prevalence and associated morbidity, treatment for Group 3 PH is limited and largely supportive with guidelines recommending targeting of the underlying pulmonary disease as the mainstay of treatment [7]. Specific targeted therapies for Group 3 PH in clinical trials has been largely influenced by successful therapies in Group 1 PAH, however when expanded to this group, results have largely been disappointing until the INCREASE trial in 2021, which resulted in the first Food and Drug Administration (FDA)-approved medication for PH associated with ILD [8]. The high disease burden and poor outcomes advocate the need for more targeted therapies in this patient population. This article will review the pathogenesis, current treatment options, and future directions based on ongoing research.

## 2. Classification

Group 3 PH is defined similarly to Group 1 PAH and formal diagnosis is made by right heart catheterization (RHC) with measurements of mean pulmonary artery pressure (PAP) > 20 mm Hg, pulmonary capillary wedge pressure (PCWP) < 15 mm Hg, and pulmonary vascular resistance (PVR) ≥ 3 wood units. These findings must be in conjunction with the presence of underlying lung disease. Group 3 PH encompasses several diseases including obstructive, restrictive, or mixed lung disease, developmental lung diseases, and includes states of chronic hypoxia such as sleep-disordered breathing, alveolar hypoventilation disorders, and chronic exposure to high altitude. This review will focus on chronic lung diseases, specifically ILDs with emphasis on idiopathic pulmonary fibrosis (IPF) and COPD. Epidemiology is difficult to define in this patient population due to differing definitions of PH and the use of transthoracic echocardiography over RHC for diagnosis. In IPF patients, evidence of PH as defined by mean PAP > 25 mm Hg ranges between 8 and 15% at initial work-up, with a higher incidence of 30–50% in severe disease, and greater than 60% in patients at end-stage disease [9,10,11,12,13,14]. Approximately 90% of severe COPD patients classified as stage IV by the GOLD criteria will have a mean PAP > 20 mmHg, yet less than 5% of patients will have a mean PAP > 35 mmHg, suggesting that severe PH occurs less frequently in COPD and that the severity of disease does not consistently correlate with the degree of PH [5,15].

## 3. Pathogenesis

The mechanisms leading to the development of Group 3 PH is not completely understood, but likely multifactorial with large contributions from the mechanisms that led to the development of underlying lung disease. Evaluations of explanted lungs of individuals with Group 3 PH demonstrate significant overlap with Group 1 PAH, suggesting a similar mechanistic process: injury to the pulmonary vascular endothelium leading to endothelial dysfunction and vascular remodeling, which in combination with sustained vasoconstriction leads to changes in pulmonary hemodynamics causing development and progression of PH.

### 3.1. Pulmonary Vascular Remodeling

Vascular remodeling refers to the structural changes in the pulmonary circulation that increases vessel wall thickness, reduces vessel lumen diameter, and thus increases PVR. The pulmonary arterial circulation is composed of the endothelial-cell-lined intima, smooth muscle cell media, and fibroblast-composed adventitia. In PAH, intermediate and large vessels demonstrate hyperplasia and hypertrophy of all three layers which occur as a result of endothelial dysfunction, characterized by disorganized hyperproliferation of pulmonary artery endothelial cells [16]. Similar changes occur in the small arterioles with the addition of plexiform vasculopathy characterized by concentric lesions consisting of endothelial and smooth muscle cells that obliterate the vessel lumen, which is a typical characteristic of PAH [17]. In patients with Group 3 PH, explanted lungs demonstrate muscularization of the microvasculature, intimal and medial proliferation, and evidence of inflammation and thrombosis. Additionally, explanted lungs demonstrate destruction of the alveoli and septa, which leads to a reduction in capillaries, contributing to increased vascular resistance [16,18,19].

### 3.2. Hypoxia

Hypoxia contributes greatly to the development of PH in this population through the alteration of pulmonary hemodynamics, inciting endothelial dysfunction, and thus giving rise to pulmonary vascular remodeling. The structural changes of the lung parenchyma and concomitant vasculature lead to aberrant gas exchange in chronic lung diseases and often results in alveolar hypoxia. Alveolar hypoxia causes a depolarization of pulmonary smooth muscle cells and an influx of cytoplasmic calcium leading to contraction and sustained pulmonary vasoconstriction. This phenomenon is specific to the lungs, as reduced oxygen delivery in the systemic circulation leads to vasodilatation. Brief hypoxia exposures lead to prolonged increases in pulmonary vascular resistance with long-term exposure leading to structural changes of the pulmonary arterioles predominantly defined as medial hypertrophy [20].

### 3.3. Interstitial Lung Disease (ILD)

ILD is a disease of progressive scarring of the lung parenchyma leading to reduced gas exchange. The structural changes as well as pathophysiologic changes associated with ILD are thought to contribute to PH development as well (Figure 1). Pulmonary fibrosis (PF) is the result of fibroblast proliferation leading to extracellular matrix deposition and obliteration of the alveoli. Fibrosis also results in damage of the pulmonary vascular bed, with decreased blood vessel density in areas of fibrosis and increased vascularization in non-fibrotic areas [21]. Fibrosis may also act to compress vessels or lead to direct remodeling, ultimately causing increased PVR. Fibrotic regions contain increased endothelial cell apoptosis, and these endothelial cells are thought to release vascular smooth muscle growth factors that lead to smooth muscle cell and fibroblast proliferation in conjunction with inflammatory mediators and oxidative stress [22,23]. Inflammation is thought to play an influential role in pathogenesis, with IPF patients having an upregulation of inflammatory mediators and gene expression demonstrating an overexpression of inflammation and hyperproliferation [24,25]. The culmination of alveolar hypoxia, endothelial dysfunction, increased oxidative stress, and inflammation all contribute to the development of PH in ILD.

### 3.4. Chronic Obstructive Pulmonary Disease (COPD) and Emphysema

COPD is a disease of airflow limitation that results in structural changes of the lung parenchyma and concomitant vasculature that leads to aberrant gas exchange. The development of PH in COPD is a result of both structural and functional causes that are inherent to chronic lung disease (Figure 2). Alveolar hypoxia contributes to pulmonary vasoconstriction, a lung-specific phenomenon. Hypoxia leads to pulmonary artery smooth muscle cell contraction, and even brief hypoxia exposures lead to prolonged increases in pulmonary vascular resistance with long-term exposure leading to structural changes of the pulmonary arterioles [20]. Individuals with COPD are not only prone to hypoxic pulmonary vasoconstriction, but also experience hemodynamic effects from prolonged hypercapnia. Severe airway obstruction in these individuals leads to alveolar hypoventilation and resultant hypercapnia. Prior studies have demonstrated a direct relationship between partial pressure of carbon dioxide (PaCO_2_) and mean PAP in COPD patients [26]. There is also a relationship between hypercapnia leading to increased cardiac output and this relationship can be explained through a few mechanisms [27]. Increased CO_2_ at the renal tubules leads to retained sodium and fluid through the sodium–hydrogen exchange. Additional fluid retention may occur through the vasodilatory effects of hypercapnia in the systemic circulation leading to neurohormonal activation of salt and water retention [28,29]. Fluid retention and edema increases pulmonary venous return, increasing stroke volume and ultimately cardiac output. Although this would lead to increased flow in the pulmonary circulation, this is also complicated by acidemia-induced pulmonary vasoconstriction [30].

The structural changes that occur due to COPD also contribute to altered pulmonary hemodynamics. Emphysema is characterized by destruction of the alveoli and associated pulmonary microvasculature; the reduction in overall vascular cross-sectional area leads to increases in pulmonary vascular pressure [16]. Additionally, the gas-trapping that occurs due to airway obstruction results in lung hyperinflation which can theoretically lead to compression of pulmonary vessels and increase pulmonary vascular pressure. These structural changes altering the pulmonary hemodynamics contribute to the development of pulmonary artery muscularization and hypertrophy contributing to further elevations in PVR [17]. Lastly, chronic smoke exposure and airway inflammation leads to medial hypertrophy; evidence of vascular remodeling is seen in smokers before the development of COPD or PH [31].

## 4. Group 3 Pulmonary Hypertension Therapies

Approach to treatment of Group 3 PH patients is centered around guideline-directed therapy of the underlying lung disease and treatment of comorbid conditions that may also exacerbate PH, such as left-sided heart disease, sleep disordered breathing, and pulmonary thromboembolism [7,32]. Patients with hypoxemia should also receive long-term oxygen therapy (LTOT). A prospective study of LTOT in COPD patients improved mean PAP and prevented worsening of PH. This recommendation is applied to other chronic lung diseases; however, no studies have addressed the benefit beyond COPD [33]. Supportive therapies such as pulmonary rehabilitation should be a part of standard therapy as well as the use of diuretics in patients that have evidence of hypervolemia and heart failure. All preventative measures should be taken to avoid exacerbations of underlying lung disease including vaccination to prevent respiratory infections and assistance in smoking cessation. Patients with advanced lung disease should be referred for transplantation when appropriate.

### 4.1. Pulmonary-Arterial-Hypertension-Specific Therapies

Treatment of PAH has largely been established by targeting vasodilator pathways of prostacyclin, endothelin-1, and nitric oxide (NO). All classes of PAH medications have been tested in patients with PH associated with ILD and COPD, yet with varying degrees of success (Table 1).

#### 4.1.1. Nitric Oxide (NO) Pathway

NO is a potent vasodilator that is synthesized in pulmonary vascular endothelial cells and functions in conjunction with other vasodilators and constrictors to maintain vascular tone in response to stress or oxygen levels [34]. In individuals with IPF, COPD, or other diseases of the pulmonary vasculature, NO production is decreased, ultimately resulting in a vasoconstrictive phenotype and impaired gas exchange [35,36]. Phosphodiesterase-5 (PDE-5) degrades cyclic guanosine monophosphate (cGMP), which is the product of NO production that acts directly on smooth muscle cells to induce vasodilatation. Inhibitors of PDE-5, such as sildenafil and tadalafil, stabilize and increase cGMP levels, favoring a vasodilatory phenotype. 

Sildenafil treatment in IPF patients demonstrated preferential vasodilatation in well-ventilated areas of the lung, improving gas exchange [37]. These favorable findings led to the STEP-IPF study which ultimately did not meet the primary outcome of improvement in six-minute walk distance (6MWD) in patients with advanced IPF but did have some positive secondary outcomes with improvement in perceived dyspnea and quality of life [38]. Another randomized control trial evaluated the effect of sildenafil in addition to the anti-fibrotic medication pirfenidone in patients with advanced IPF and increased risk of PH, defined as mPAP ≥ 20 mm Hg with PCWP < 15 mm Hg or evidence of intermediate/high probability PH on echocardiogram. The authors hoped that earlier targeted PH therapy in patients with advanced IPF would be an ideal approach as vascular changes exist prior to definitive development of PH. The addition of sildenafil compared to pirfenidone monotherapy did not meet the primary endpoint of disease progression measured by 6MWD, respiratory-associated hospitalization, or all-cause mortality [39]. 

Riociguat is a soluble guanylate cyclase stimulator that acts similarly through the NO pathway to increase production of cGMP and has been identified as a successful therapy in both primary PAH and CTEPH [40,41]. Preclinical models have demonstrated antifibrotic effects, supporting its role as a potential therapeutic for ILD-related PH [42,43]. The RISE-IIP study evaluated riociguat treatment in idiopathic interstitial pneumonia with precapillary PH and failed to show improvement in 6MWD, but was also associated with serious adverse events, increased mortality, and therefore the study was terminated early [44]. Pulse inhaled NO is currently being evaluated as a potential therapeutic, and early clinical studies suggest that IPF patients at risk of PH have improved physical activity with this therapy [45]. The REBUILD study is currently underway to further evaluate pulse inhaled NO in PH associated with IPF patients who are on LTOT [46].

There have been more studies evaluating NO-targeted therapy in COPD patients, however results are inconclusive across trials. In patients with COPD and without PH, two studies of sildenafil showed opposing results, one with improvement in exercise capacity while sildenafil was associated with increased harm in the other [47,48]. Blanco et al. conducted a study that demonstrated improvement in hemodynamics with sildenafil in COPD-associated PH, however decreased arterial oxygenation at rest was found and attributed to sildenafil inhibiting pulmonary vasoconstriction that may occur in response to hypoxia, leading to worsening gas exchange from ventilation/perfusion (V/Q) mismatching [49]. Subsequent randomized controlled trials have not demonstrated any significant worsening of oxygenation from PDE-5 inhibitor use in this population. Sildenafil, in addition to pulmonary rehabilitation in patients with severe COPD and moderate PH, did not improve exercise capacity and neither did tadalafil in patients with COPD and mild PH [50,51]. Lastly, in a pilot study of patients with severe PH and COPD, sildenafil treatment improved PVR and other secondary endpoints including the BODE (body mass, airflow obstruction, dyspnea, exercise capacity) index [52]. The differing results and the inconsistent study population across severity of PH and COPD make the data difficult to interpret and therefore PDE-5 inhibitors are not recommended in Group 3 PH guidelines [32,53].

#### 4.1.2. Endothelin (ET) Pathway

Endothelin-1 (ET-1) is a potent vasoconstrictor peptide produced predominately by endothelial cells. ET-1 acts via both autocrine and paracrine signaling on vascular endothelial and smooth muscle cells, mediated by ETA and ETB receptors to execute its vasoconstrictive and mitogenic functions. ET-1 activation of ETA and ETB receptors on smooth muscle cells leads to vasoconstriction, whereas activation of ETB receptors on endothelial cells stimulates production of vasodilatory compounds, NO and prostaglandin, and aids in pulmonary clearance of ET-1, thus the receptors are thought to function in mediating vasomotor tone [54]. ET-1 concentration in healthy individuals is low, but elevated in patients with PAH, IPF, and COPD, suggesting that activation of the ET pathway greatly contributes to disease [55,56]. ET receptors are also present in fibroblasts, leading to increased collagen and fibrosis formation, making this an ideal treatment target for pulmonary vascular disease. Ambristentan is a selective ETA receptor antagonist, whereas bosentan is a dual ETA and ETB receptor antagonist; they are both approved treatments for Group 1 PAH. 

As ET-1 influences fibroblast proliferation and inflammation, ET receptor antagonists have been tested in patients with IPF and PH. Ambistentan treatment in this population was not effective in treating disease progression, led to harm causing early termination of the ARTEMIS-IPF study, and is contraindicated in this patient population [57]. Bosentan, the dual receptor antagonist, similarly did not show improvement in hemodynamics or functional capacity in patients with PH and fibrotic idiopathic interstitial pneumonia [58].

**Table 1 pharmaceuticals-16-00418-t001:** Randomized, controlled trials in pulmonary hypertension associated with chronic lung disease.

Trial Study	Therapy	Target	Outcome	Ref
**Pulmonary Fibrosis-Associated Pulmonary Hypertension (PF-PH)**				
STEP-IPF Zisman et al., 2010	Sildenafil	NO	No improvement in 6MWD	[38]
Behr et al., 2021	Sildenafil and Pirfenidone	NO	No improvement in 6MWD, respiratory hospitalization, or mortality	[39]
RISE-IIP Nathan et al., 2019	Riociguat	NO	No improvement in 6MWD; increased adverse events and mortality	[44]
iNO-PF Nathan et al., 2020	Pulsed inhaled NO	NO	Increased moderate/vigorous physical activity	[45]
ARTEMIS-IPF Raghu et al., 2013	Ambrisentan	ET-1	No improvement in lung function, respiratory hospitalization, or death; Increased harm	[57]
BPHIT Corte et al., 2014	Bosentan	ET-1	No decrease to PVR index of 20% or more	[58]
INCREASE Waxman et al., 2021	Inhaled Treprostinil	Prostacyclin	Improvement in 6MWD	[8]
**Chronic Obstructive Pulmonary Disease-Associated Pulmonary Hypertension (COPD-PH)**				
Blanco et al., 2010	Sildenafil	NO	Reduced mean PAP	[49]
Blanco et al., 2013	Sildenafil and pulmonary rehabilitation	NO	No improvement in cycle endurance time	[50]
Goudie et al., 2014	Tadalafil	NO	No improvement in 6MWD	[51]
SPHERIC-1 Vitulo et al., 2017	Sildenafil	NO	Reduced PVR	[52]
Stolz et al., 2008	Bosentan	ET-1	No improvement in 6MWD	[59]
Valerio et al., 2009	Bosentan	ET-1	Reduced mean PAP and PVR, Increased 6MWD, and reduced BODE index	[60]

Ref indicates reference; NO, nitric oxide; 6MWD, six-minute walk distance; ET-1, endothelin-1; PVR, pulmonary vascular resistance; PAP, pulmonary artery pressure; BODE, body mass index, airflow obstruction, dyspnea, and exercise performance measure.

Bosentan treatment in severe COPD without severe PH did not improve pulmonary hemodynamics or lung function, but also worsened hypoxemia and functional status in this patient population [59]. Another study looking at bosentan in COPD-PH showed improvement in hemodynamics and BODE index [60]. Conflicting results, small sample sizes, and concern for harm suggest that endothelin receptor antagonists should not be used for patients with Group 3 PH.

#### 4.1.3. Prostacyclin Pathway

Prostacyclin is an endogenous vasodilator produced by vascular endothelial cells. Prostacyclin binds to a G-protein-coupled receptor and results in downstream production of cyclic adenosine monophosphate (cAMP), leading to vasodilatation of vascular smooth muscle cells [61]. Through an increase in intracellular cAMP, prostacyclin also functions to inhibit platelet aggregation. 

Synthetic prostacyclin analogues have been developed in several formulations and function to cause direct vasodilatation of vascular beds and are approved for treatment of PAH. The INCREASE trial recently demonstrated that inhaled Treprostinil led to improvement in exercise capacity measured by 6MWD and reduced NT-pro-BNP levels and disease-related exacerbations in individuals with IPF and PH. Investigators note that inhaled administration allows for preferential blood flow to well-ventilated alveoli and minimizes V/Q mismatching from vasodilators in Group 3 patients [8]. Subsequent analysis also demonstrated significant improvement in forced vital capacity (FVC), leading to the TETON trial which is evaluating the effect of inhaled Treprostinil in IPF on FVC. Inhaled Treprostinil is the only FDA-approved PAH therapy for treatment of Group 3 PH associated with ILD. Trials are also currently underway on patients with PH related to COPD [62].

### 4.2. Emerging Molecular Targets for Pulmonary Hypertension Related to Chronic Lung Disease

As discussed previously, clinical trials for chronic lung disease have largely been undertaken to repurpose therapies that are efficacious in PAH by targeting one of the vasodilator pathways: prostacyclin, endothelin-1, and NO. In PAH, these therapies slow progression of disease, but were largely ineffective in patients with Group 3 PH until the recent INCREASE trial [8,38,39,44,48,49,50,51,59,63]. Since the development of PH is a large predictor of mortality in individuals with chronic lung disease, effective therapeutics that target and reverse vascular remodeling are needed. The following section discusses emerging molecular targets that have demonstrated consistent success in preclinical and early clinical models of Group 3 PH (Figure 3). Larger bodies of research have been established in the ILD model of Group 3 PH as this likely reflects the higher burden of PH incidence within this group and the well-established bleomycin murine model of IPF and secondary PH (Table 2). Molecular targets for both ILD and COPD will be addressed jointly.

#### 4.2.1. Bone Morphogenic Protein Receptor Type II (BMPR2)

Heterozygous BMPR2 mutations are implicated in 70–80% of heritable PAH cases and 10–20% of idiopathic PAH cases [64]. This mutation results in reduced function of BMPR2, a member of the transforming growth factor-beta (TGF-β) superfamily, and subsequent loss of downstream signaling. BMPR2 aids in the regulation and suppression of TGF-β which functions across several cell types modulating cell growth and differentiation [65]. BMPR2 mutations have been associated with the pulmonary-artery-hyperproliferative and apoptosis-resistant phenotype that leads to vascular remodeling; however, there is also a reduction in BMPR2 expression in other forms of PAH not associated with clear mutations [66]. Chen and colleagues found that BMPR2 expression and signaling are decreased in lung tissue of IPF patients with and without PH, which also correlates with the severity of PH. Reduced BMPR2 expression is also seen in macrophages of bleomycin-treated mice with the development of vascular remodeling and PH that is mediated by interleukin-6 (IL-6) and microRNAs targeting BMPR2 degradation [67]. In another study evaluating mice expressing mutant BMPR2, bleomycin exposure resulted in more severe PH with increased HIF1-α expression [68]. Additionally, treatment with recombinant BMP9 reverses bleomycin-induced PH by restoring BMPR2 and SMAD signaling pathways [69]. Targeting BMPR2 and TGF-β has preliminarily been a promising therapeutic approach in recent clinical trials for PAH patients. Sotatercept is a ligand trap that is specific for the TGF-β family and allows for a rebalancing of proliferative and anti-proliferative signaling that is offset by reduced BMPR2 function [70]. In a Phase 2 clinical trial, sotatercept was well tolerated and demonstrated a significant reduction in PVR in PAH patients [71]. The improvement in PVR is attributed to a reduction in mean PAP without a change in cardiac output or PCWP, suggesting that the therapeutic effect is likely a result of reduced vascular remodeling as opposed to a vasodilatory effect. As pulmonary vasodilators have a limited role in PH associated with chronic lung disease and preclinical models support a role of BMPR2 signaling in PH and fibrosis pathobiology, this therapeutic warrants expanded exploration in Group 3 patients.

#### 4.2.2. Angiogenesis and Vascular Endothelial Growth Factor (VEGF)

Pulmonary fibrosis not only causes obliteration of the alveoli, but also damages the pulmonary vasculature. Angiogenesis is the process of new vessel formation from existing vasculature and is an important mechanism after tissue damage to facilitate healing. The abundance or lack of pulmonary capillaries is thought to influence the development of PH in PF. In explanted lungs of IPF, vessel density is decreased in areas of fibrosis and increased in non-fibrotic areas, but overall results in reduced vessel density [21]. Additionally, VEGF levels are reduced in explanted IPF lungs with an upregulation of angiostatic molecule pigment epithelium-derived factor, suggesting that an imbalance of angiogenic factors may be contributing to pathology [72,73]. VEGF is abundant in the lungs and important for maintenance of the pulmonary endothelium, contributing to both NO and prostacyclin production. Animal models of chronic hypoxia express increased VEGF levels and development of vascular remodeling and PH [74,75,76]. However, the frequently studied Sugen-hypoxia animal model that closely mimics PAH with the development of vascular remodeling and plexiform lesions uses a VEGF receptor (VEGFR2) antagonist in conjunction with hypoxia, which is known to upregulate VEGF levels, demonstrating that VEGF signaling is important for the development of PH. A PF rat model of adenoviral delivery of TGF-β1 led to the development of PH with increased vascular apoptosis and increased VEGF expression in highly fibrotic regions [77]. VEGF treatment attenuated these findings but did lead to a worsening of fibrosis. VEGFR2 modulates a survival pathway and downregulation leads to increased PF and decreased vascular density, contributing to PH development in PF. Taken together, this suggests that VEGF is important to the development of PH and the modulation of VEGF balance at certain time points in the disease process could be an important therapeutic target.

#### 4.2.3. Nuclear Factor-Kappa B (NF-κB) Signaling and Oxidative Stress

Inflammatory responses to environmental triggers are mediated by NF-κB signaling and thus activation of NF-κB is thought to play a role in the pathogenesis of asthma, COPD, and autoimmune diseases [78,79,80,81]. NF-κB signaling is involved in angiogenesis and vascular cell proliferation, upregulated in lung tissue from IPAH patients, and its inhibition attenuates PAH development in monocrotaline mouse models [82,83,84]. Patients with COPD-associated PH have increased serum TLR-4, an NF-κB upstream activator, and NF-κB expression correlates with PH severity [85]. Additionally, reduction in NF-κB signaling in a COPD-PH mouse model (intratracheal-elastase-induced emphysema) through budesonide/glycopyrronium/formoterol fumarate triple therapy prevented PH development and decreased COPD progression [86].

Nuclear factor erythroid 2-related factor 2 (Nrf2) is a transcription factor that regulates NF-κB signaling, reactive oxygen species, and overall generation of oxidative stress. Treatment with Nrf2 induction in a chronic hypoxia PAH murine model decreased vascular remodeling and RV hypertrophy [87]. Bardoxolone methyl is an Nrf2 activator that decreases pro-inflammatory NF-κB and reduces oxidative stress. The LARIAT phase 2 clinical trial used bardoxolone methyl as treatment in both PAH and PH related to chronic lung disease and demonstrated improvements in exercise capacity, specifically in connective-tissue-disease-associated PAH. These findings prompted the CATALYST and RANGER phase 3 clinical trials, which were delayed due to the COVID-19 pandemic, but ultimately are halted as data suggested that the primary endpoint, improved 6MWD, would not be reached [88]. The combination of NF-κB signaling and reduction in ROS makes this an ideal potential therapy for PH related to chronic lung disease, as preclinical models have also suggested ROS reduction as an attenuator of vascular remodeling and PH in models of chronic hypoxia and pulmonary fibrosis [89,90,91].

#### 4.2.4. Pulmonary Renin–Angiotensin System (RAS)

RAS is a group of ligands and receptors that regulate several organ systems and has been a well-established therapeutic target for several disease processes. Activation of angiotensin II (Ang II), a potent vasoconstrictor, stimulates lung fibroblast growth, upregulates TGF-β, and stimulates pulmonary artery smooth muscle cell growth contributing to pathophysiology seen in both PF and PH [92,93,94]. These detrimental processes are stimulated by angiotensin-converting enzyme (ACE). ACE2 is a homolog of ACE that provides an alternative pathway for pulmonary RAS regulation with the vasodilatory end-product angiotensin-(1-7) that counteracts the negative effects of Ang-II and leads to its degradation. ACE2 treatment has been effective in improving PH and reducing vascular remodeling in bleomycin murine models of IPF and monocrotaline model of PAH [95,96,97]. Additionally, the Ang-(1-7) pathway has increased expression of the angisotensin type 2 (AT2) receptor and stimulation of this also improved lung fibrosis and vascular remodeling in a preclinical PF-PH murine model [97,98]. Patients with PAH have reduced ACE2 expression and a pilot study demonstrated that recombinant protein infusion improved PVR and cardiac output and reduced oxidative stress and inflammatory mediators; a phase-2 trial is currently recruiting for PAH patients [99,100].

#### 4.2.5. Peroxisome Proliferator-Activated Receptors (PPAR)

Peroxisome proliferator-activated receptors (PPARs) are ligand-activated transcription factors belonging to the nuclear receptor family that regulates inflammation and lipid and carbohydrate metabolism. These receptors are ubiquitous across many cell types and PPARγ is present predominantly in adipose tissue, but also found in lung parenchyma and immune cells with several pulmonary diseases, including PH demonstrating reduced PPARγ expression [101,102,103,104]. PPARγ regulates cytokines involved in PH and vasoconstrictors including endothelin-1, and overall contributes to key PH mechanisms including proliferation, inflammation, apoptosis, and angiogenesis of pulmonary vascular cells [105]. Further studies have supported a role for PPARγ in reversing vascular remodeling in preclinical PAH models [106] and suggest that this mediation occurs via inhibition of TGF-β1 signaling [107]. 

In pulmonary fibrosis, tissue injury through proinflammatory and abnormal growth factor production leads to the transition of mesenchymal cells to myofibroblasts which produce collagen and extracellular matrix proteins. PPARs are shown to have anti-fibrotic effects and decrease lung fibrosis in preclinical models [108,109,110]. Lanifibranor (IVA337) is a pan-PPAR agonist and has shown success in the treatment of liver fibrosis [111]. Lanifibranor has demonstrated success with pulmonary fibrosis in preclinical murine models of IPF, non-specific interstitial pneumonia (NSIP), and systemic sclerosis, with some improvement in secondary PH development [112,113]. Furthermore, lanifibranor inhibited human fibroblast to myofibroblast transition and proliferation mediated by TGF-β [112].

Taken together, PPAR activation has demonstrated promising pre-clinical results and its synthetic ligands are well-studied therapeutics for a wide range of disease processes. Additionally, PPARγ’s therapeutic benefit has significant overlap with several of the previously discussed targets in this section, including BMPR2 and TGF-β signaling [105,114,115] and the RAS system as Ang II inhibition increased PPARγ expression in murine models of renal fibrosis [116].

#### 4.2.6. Endothelial to Mesenchymal Transition (EndoMT)

EndoMT is a process of cellular transdifferentiation where endothelial cells lose their barrier protective functions and gain mesenchymal characteristics such as contractility [117]. EndoMT has been implicated in pulmonary artery intima proliferation and remodeling in PAH models [118] and evidence of EndoMT has been found in patients with PAH and PH associated with IPF and systemic sclerosis [119,120]. EndoMT in the pathophysiology of vascular remodeling in PH is induced by several signaling pathways and mediators. Specifically, in preclinical murine models of PH associated with PF, PH and vascular remodeling have been improved by inhibition of JAK2/SMAD3, a known regulator of endothelial cell injury, and ERK 1/2 signaling through sildenafil treatment and direct inhibition of JAK2 or interleukin-11 (IL-11) [120,121,122]. The activation of myofibroblasts is integral to the pathobiology of lung fibrosis and may also contribute to vascular remodeling. These preclinical studies also identify EndoMT as a source of myofibroblasts and pulmonary artery smooth muscle cell transition to a myofibroblast phenotype as a contributing process [120,122,123]. 

Dipeptidyl peptidase IV (DPP-4) is a serine protease that cleaves N-terminal dipeptides from substrates and is increased in inflammatory disease states. DPP-4 inhibition (DPP-4i) has been used in the treatment of diabetes and cardiovascular complications. It also has been shown to regulate endothelial and smooth muscle vascular cells. DPP4i attenuated pulmonary fibrosis in a sepsis lung murine model by decreasing the EndoMT process [124]. DPP-4i treatment inhibits EndoMT and pulmonary vascular remodeling in a monocrotaline PAH model and reduced PH in bleomycin and chronic hypoxia mouse models [125]. DPP-4i is a well-established therapeutic approach and shows promise in the treatment of PH and vascular remodeling through its protective effects on the pulmonary endothelium from EndoMT.

#### 4.2.7. Hypoxia–Adrenergic Axis

In response to cellular injury, adenosine acts via extracellular signaling to regulate tissue repair. In chronic lung disease, elevated levels of adenosine are thought to contribute to remodeling and progression of disease, with findings of increased adenosine 2B receptor expression in patients with COPD and IPF [126,127]. Nucleotide adenosine has been upregulated in lung tissue remodeling and known fibroproliferative mediators, such as IL-6, ET-1, and reactive oxygen species, are regulated through the adenosine signaling system. Karmouty-Quintana and colleagues found that selective antagonism (GS-6201) or deletion of the adenosine 2B receptor prevented vascular remodeling and PH, reduced lung fibrosis, and downregulated IL-6 and ET-1 in mice exposed to bleomycin [128]. Upregulation of inflammatory cells, specifically activated macrophages, is known to play a role in PF and PH. The same group evaluated adenosine receptors in myeloid cells, and findings suggested that the adenosine receptor ADORA2B is increased in macrophages from IPF patients and preclinical models, and ADORA2B antagonism improves fibrosis and PH in mouse models. In bleomycin-treated mice, deletion of myeloid cell ADORA2B led to alteration in macrophages and decreased lung fibrosis and fibrotic mediators in addition to reduced pulmonary remodeling and PH [129]. Hypoxia-inducible factor (HIF) is stabilized in hypoxic conditions and is known to play a role in several chronic lung disease processes in addition to development of PH. HIF1α is known to activate adenosine signaling, noted by increased expression of enzymes that aid in the synthesis of adenosine and increased expression of adenosine receptor 2B. In patients with IPF and PH, HIF1α is stabilized by decreased mitochondrial metabolism leading to impaired succinate metabolism when compared to IPF patients alone. This stabilization of HIF1α leads to the enhancement of adenosine signaling and increased use of adenosine receptors [130]. 

In the realm of COPD, a mouse model of airspace enlargement as seen in emphysema, is generated by adenosine deaminase (ADA) deficiency. ADA is an enzyme that metabolizes adenosine, and ADA is reduced in patients with COPD. Genetic deficiencies of ADA in mice leads to increased adenosine expression, airspace enlargement, and development of vascular remodeling and PH. Patients with COPD and PH have remodeled pulmonary vessels with smooth muscle and collagen deposition, and increased levels of ADORA2B correlate with PAP. Treatment of ADORA2B antagonist in this ADA-deficient COPD mouse model led to the attenuation of PH through a regulation of hyaluranon and hyaluranon synthase-2, suggesting that adenosine signaling influences lung extracellular matrix composition which leads to remodeling [128].

#### 4.2.8. Hypoxia-Inducible Factor

Hypoxia-inducible factor signaling is known to be important in the development of PAH. HIF is a transcription factor that degrades in normoxia, but in response to low oxygen levels translocates to the nucleus and augments gene transcription. Although the mechanisms that result in fibrosis and lead to vascular remodeling are unclear, stabilization of HIF1α under hypoxic conditions could play a crucial role as hypoxia is an important feature of the disease. Studies have also suggested a potential dysregulation in oxygen sensing as HIF is often upregulated in normoxic conditions. Additionally, individuals exposed to hypoxia demonstrate a wide change in mean PAP, suggesting additional influencing factors in the development of PH related to hypoxia [131]. Interestingly endothelial-HIF-deficient mice exposed to bleomycin do not show significant differences in lung fibrosis, but do show protection against the development of PH and vascular remodeling, suggesting that HIF may have an independent role from PF in PH pathobiology [72,132].

**Table 2 pharmaceuticals-16-00418-t002:** Emerging therapies in pulmonary hypertension associated with chronic lung disease.

Intervention	Target	Model Studied	Outcome	Ref
**Pulmonary Fibrosis-Associated Pulmonary Hypertension (PF-PH)**				
IL-6 -/- or soluble GP130 (IL-6 inhibitor)	BMPR2	Bleomycin mice	Increased BMPR2 expression; abrogated development of PH; reduced development of PF	[67]
Recombinant BMP9	BMPR2	Bleomycin rat	Restored BMPR2 signaling; prevents bleomycin-induced PH and PF	[69]
Adenoviral delivery of VEGF	VEGF	Adenoviral delivery of TGFβ-1 in rats	Reduced PAP and pulmonary vascular remodeling; worsened PF	[77]
Ang-(1-7) or ACE2 overexpression	RAS	Bleomycin rat Monocrotaline rat	Prevented PH and PF	[95]
Recombinant ACE2	RAS	Bleomycin mice	Attenuated pulmonary vascular remodeling	[96]
Compound 21 (AT2 receptor agonist)	RAS	Bleomycin rat	Reduced progression of PF, PH, and muscularization of pulmonary vessels	[98]
IVA337 (pan-PPAR agonist)	PPAR	Bleomycin mice Fra-2 transgenic mice	Prevented PF development; improves PH and vascular remodeling in Fra-2 transgenic mice	[112]
siRNA IL-11	EndoMT	Bleomycin mice IL-11 treated mice	Attenuated PF, PH, and vascular remodeling; reduces evidence of EndoMT	[122]
Sitagliptin (DPP4 inhibitor)	EndoMT	MCT rat Bleomycin rat Chronic hypoxia rat	Attenuated PH, RV and pulmonary vascular remodeling, and EndoMT in MCT rats; prevented PH in bleomycin and chronic hypoxia rats	[125]
ADORA2B myeloid cell KO	Hypoxia-adrenergic axis	Bleomycin mice	Attenuated PF, improved lung function, and prevented PH	[129]
Endothelial HIF deficiency	HIF	Bleomycin mice	Prevented PH and RV and vascular remodeling	[132]
**Chronic Obstructive Pulmonary Disease-Associated Pulmonary Hypertension (COPD-PH)**				
Budesonide glycopyrronium formoterol fumarate therapy	NF-κB	Intratracheal elastase induced emphysema	Prevented PH development and COPD progression	[86]
ADORA2B blockade	Hypoxia-adrenergic axis	Adenosine deaminase deficient mice	Attenuated development of PH	[128]

Ref indicates reference; BMPR2, bone morphogenetic protein receptor 2; VEGF, vascular endothelial growth factor; TGFβ, transforming growth factor β; PAP, pulmonary artery pressure; RAS, renin-angiotensin system; PPAR, peroxisome proliferator-activated receptor; EndoMT, endothelial to mesenchymal transition; HIF, hypoxia-inducible factor; RV, right ventricle; NF-κB, nuclear factor-kappa B.

## 5. Conclusions

Pulmonary hypertension is an important and deadly complication of chronic lung diseases, including ILD and COPD. There are both overlapping and distinct pathogenic mechanisms associated with the development of PH in these conditions. It remains imperative that we use the best standard of care to treat the underlying respiratory condition, yet the severe pathobiologic changes in the pulmonary circulation associated with the development of PH cannot be ignored. Vasoconstriction and vascular remodeling continue to be the areas that are considered to be the most ripe for targeted therapeutics; yet to date, only inhaled treprostinil has attained FDA approval for the treatment of PH associated with ILD and no targeted therapies are available for PH associated with COPD. We have reviewed many emerging therapeutic targets based on important molecular mechanisms in PH associated with both ILD and COPD, which include BMPR2, angiogenesis, NF-κB, RAS, PPARγ, EndoMT, adrenergic stimuli, and HIF signaling. While our understanding of these molecular targets and signaling pathways in Group 3 PH has grown exponentially, much of the work has remained in the preclinical stages of investigation. Continuing to develop our understanding of PH related to ILD and COPD, as well as bringing novel, targeted therapies into development and clinical studies should remain a priority with this deadly condition.

## Figures and Tables

**Figure 1 pharmaceuticals-16-00418-f001:**
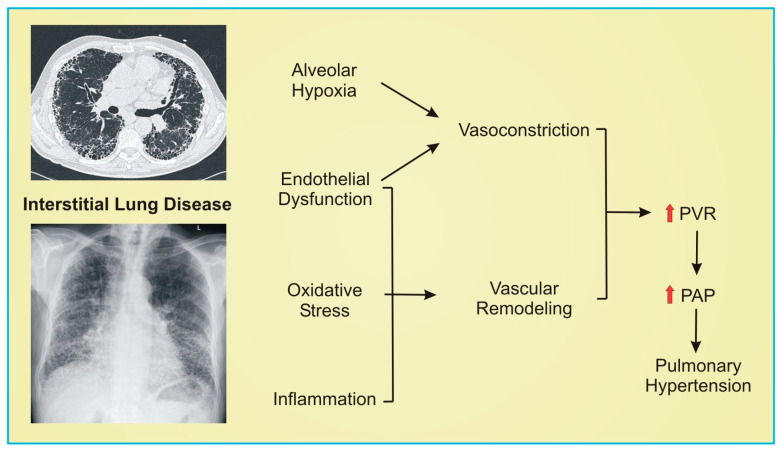
Schematic representation of pathogenesis of pulmonary hypertension in interstitial lung disease. Axial computed tomography image of a patient with fibrotic lung disease with significant traction bronchiectasis, subpleural honeycombing, and fibrosis (upper left image). Frontal chest radiograph demonstrating bilateral lower lobe fibrosis (lower left image). Red arrows indicate increase. PVR indicates pulmonary vascular remodeling; PAP, pulmonary artery pressure.

**Figure 2 pharmaceuticals-16-00418-f002:**
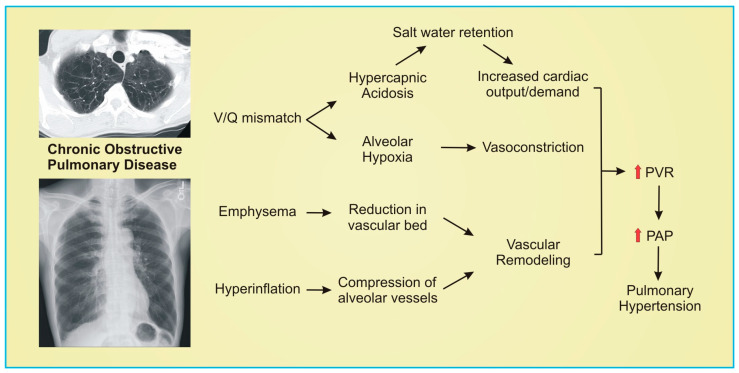
Schematic representation of pathogenesis of pulmonary hypertension in chronic obstructive pulmonary disease. Axial computed tomography image of a patient with bilateral upper lobe emphysema (upper left image). Frontal chest radiograph demonstrating lung hyperinflation (lower left image). Red arrows indicate increase. PVR indicates pulmonary vascular resistance; PAP, pulmonary arterial pressure.

**Figure 3 pharmaceuticals-16-00418-f003:**
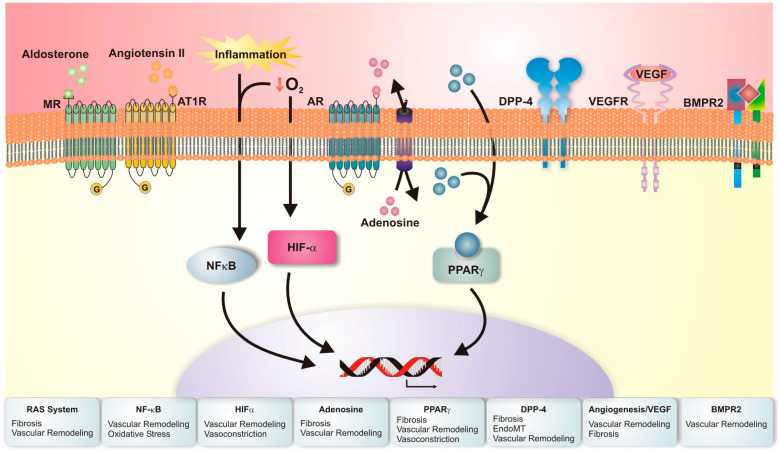
Molecular mechanisms associated with Group 3 pulmonary hypertension. This schematic figure summarizes the molecular mechanisms associated with emerging therapies in preclinical and early clinical development for Group 3 pulmonary hypertension. At the bottom of the figure the pathobiologic changes associated with each molecular mechanism are listed. MR indicates mineralocorticoid receptor; AT1R, angiotensin 1 receptor; O_2_, oxygen; AR, adenosine receptor; DPP-4, dipeptidyl peptidase type 4; VEGFR, vascular endothelial growth factor receptor; VEGF, vascular endothelial growth factor; BMPR2, bone morphogenetic protein receptor 2; NFκB, nuclear factor-kappa b; HIF-α, hypoxia-inducible factor alpha; PPARγ, peroxisome proliferator-activated receptor gamma; RAS, renin-angiotensin system.

## Data Availability

Not applicable.

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
