# Peer review of "Pharmacology and Emerging Therapies for Group 3 Pulmonary Hypertension Due to Chronic Lung Disease"

_pharmaceuticals, 2023, doi:10.3390/ph16030418_

Round 1
Reviewer 1 Report
The authors wrote an interesting review about Pulmonary hypertension. Authors wrote that vasoconstriction and vascular remodeling continue to be the areas that are considered to be the most ripe for targeted therapeutics, but they did not gave the information about endothelial cells in Pulmonary hypertension. They must add it. It is known that endothelial cells participates in vascular remodeling.
Author Response
We would like to thank you for taking the time to read our manuscript. We sincerely appreciate all of your enthusiasm on the study and your helpful, constructive comments on the manuscript. We hope the revision continues to provide a satisfactory response.
Reviewers Point 1: Authors wrote that vasoconstriction and vascular remodeling continue to be the areas that are considered to be the most ripe for targeted therapeutics, but they did not gave the information about endothelial cells in Pulmonary hypertension. They must add it. It is known that endothelial cells participates in vascular remodeling.
Response: Thank you for highlighting the importance of endothelial dysfunction in the pathogenesis of PH. We agree that it is an important mechanism and we have added a line under the pulmonary vascular remodeling section to highlight this. It is also discussed further as contributing to pathogenesis in the IPF pathogenesis section and is included in Figure 1. The emerging therapies section also discusses the importance of endothelial dysfunction in relation to specifically described targets such as BMPR2 and EndoMT, all of which contribute to vascular remodeling, conveying the importance of endothelial dysfunction in pulmonary vascular remodeling.
Reviewer 2 Report
The manuscript is well-written and the authors did a good job in summarizing and presenting this interesting topic.
Author Response
Reviewers Point 1: The manuscript is well-written and the authors did a good job in summarizing and presenting this interesting topic.
Response: We would like to thank you for taking the time to read our manuscript. We sincerely appreciate all of your enthusiasm on the study and your helpful, constructive comments on the manuscript. We hope the revision continues to provide a satisfactory response.
Reviewer 3 Report
The manuscript entitled “Pharmacology and Emerging Therapies for Group 3 Pulmonary Hypertension Due to Chronic Lung Disease” represents a systematic review of pathophysiology and therapeutic targets regarding group 3 pulmonary hypertension. The authors covered the preclinical and clinical trials related to molecular targets. One suggestion-the authors should consider adding the additional table (with references from the manuscript). One where the most important animal studies (species, vascular and pulmonary changes, etc.) and molecular targets would be linked and highlighted, and another similar where human studies (gender, age characteristics, etc.) would be included.
Author Response
We would like to thank you for taking the time to read our manuscript. We sincerely appreciate all of your enthusiasm on the study and your helpful, constructive comments on the manuscript. We hope the revision continues to provide a satisfactory response.
Reviewers Point 1: One suggestion-the authors should consider adding the additional table (with references from the manuscript). One where the most important animal studies (species, vascular and pulmonary changes, etc.) and molecular targets would be linked and highlighted, and another similar where human studies (gender, age characteristics, etc.) would be included.
Response: Thank you for suggesting table format to better visually present the clinical and preclinical trails discussed. We agree that for such an extensive review, a table reference makes it easier for readers to navigate highlighted studies mentioned in the review. Please see table 1 and table 2 in the revised form. Table 1 details randomized controlled trials and table 2 describes highlighted preclinical studies.
Reviewer 4 Report
1. The most common lung disease resulting in pulmonary hypertension are COPD, ILD, and obstructive sleep apnea (OSA), but is also associated with other disease, such as cystic fibrosis and high-altitude exposure. It would be better to mention that in this review manuscript.
2. Authors should list all the therapies that involved in treating PH in one table, together with their pathways, references, outcomes, phase clinical trials or marketed.
3. Authors should list all the abbreviation in one table.
4. It would be better to combine some drug/siRNA/miRNA delivery for treating PH.
Author Response
We would like to thank you for taking the time to read our manuscript. We sincerely appreciate all of your enthusiasm on the study and your helpful, constructive comments on the manuscript. We hope the revision continues to provide a satisfactory response.
Reviewers Point 1: The most common lung disease resulting in pulmonary hypertension are COPD, ILD, and obstructive sleep apnea (OSA), but is also associated with other disease, such as cystic fibrosis and high-altitude exposure. It would be better to mention that in this review manuscript.
Response 1: Thank you for noting the lack of additional disorders that give rise to group 3 PH. Under the classification section, we now list the various additional disorders, but specify that this review will focus on COPD and ILD.
Reviewers Point 2: Authors should list all the therapies that involved in treating PH in one table, together with their pathways, references, outcomes, phase clinical trials or marketed.
Response 2: Thank you for suggesting table format to better visually present the clinical and preclinical trails discussed. We agree that for such an extensive review, a table reference would make it easier for readers to navigate highlighted studies mentioned in the article. Please see table 1 and table 2 in the revised form. Table 1 details randomized controlled trials and Table 2 describes highlighted preclinical studies.
Reviewers Point 3: Authors should list all the abbreviation in one table.
Response 3: We have added an abbreviations table at the start of the manuscript as a reference for readers.
Reviewers Point 4: It would be better to combine some drug/siRNA/miRNA delivery for treating PH.
Response 4: While we agree that combination delivery would be ideal, several of these suggested mechanisms are very preclinical and likely further studies are needed to suggest combination delivery.
Reviewer 5 Report
This manuscript is well-designed and presents important current data in the field of pharmacology and therapy of pulmonary hypertension.
In my opinion, the work can be published in this form
Author Response
Reviewers Point 1: The manuscript is well-written and the authors did a good job in summarizing and presenting this interesting topic.
This manuscript is well-designed and presents important current data in the field of pharmacology and therapy of pulmonary hypertension. In my opinion, the work can be published in this form.
Response: We would like to thank you for taking the time to read our manuscript. We sincerely appreciate all of your enthusiasm on the study and your helpful, constructive comments on the manuscript. We hope the revision continues to provide a satisfactory response.